# Microwave-Induced In Situ Amorphization: A New Strategy for Tackling the Stability Issue of Amorphous Solid Dispersions

**DOI:** 10.3390/pharmaceutics12070655

**Published:** 2020-07-11

**Authors:** Wei Qiang, Korbinian Löbmann, Colin P. McCoy, Gavin P. Andrews, Min Zhao

**Affiliations:** 1School of Pharmacy, Queen’s University Belfast, Belfast BT9 7BL, UK; wqiang01@qub.ac.uk (W.Q.); C.McCoy@qub.ac.uk (C.P.M.); g.andrews@qub.ac.uk (G.P.A.); 2Department of Pharmacy, University of Copenhagen, 2100 Copenhagen, Denmark; korbinian.loebmann@sund.ku.dk; 3China Medical University- Queen’s University Belfast Joint College (CQC), China Medical University, Shenyang 110000, China

**Keywords:** microwave irradiation, amorphous solid dispersions, poorly soluble drugs, in situ amorphization, physical stability

## Abstract

The thermodynamically unstable nature of amorphous drugs has led to a persistent stability issue of amorphous solid dispersions (ASDs). Lately, microwave-induced in situ amorphization has been proposed as a promising solution to this problem, where the originally loaded crystalline drug is in situ amorphized within the final dosage form using a household microwave oven prior to oral administration. In addition to circumventing issues with physical stability, it can also simplify the problematic downstream processing of ASDs. In this review paper, we address the significance of exploring and developing this novel technology with an emphasis on systemically reviewing the currently available literature in this pharmaceutical arena and highlighting the underlying mechanisms involved in inducing in situ amorphization. Specifically, in order to achieve a high drug amorphicity, formulations should be composed of drugs with high solubility in polymers, as well as polymers with high hygroscopicity and good post-plasticized flexibility of chains. Furthermore, high microwave energy input is considered to be a desirable factor. Lastly, this review discusses challenges in the development of this technology including chemical stability, selection criteria for excipients and the dissolution performance of the microwave-induced ASDs.

## 1. Introduction

Oral administration is the most commonly used route of administration, offering various advantages including convenience of administration, compliance of medication, safety and economy. However, with the advent of combinatorial chemistry and high-throughput screening technology, a significant number of bioactive lipophilic and hydrophobic compounds have been synthesized but exhibit poor aqueous solubility and concomitantly poor oral bioavailability [1]. Currently, it is estimated that about 90% of leading compounds in the drug discovery process, 75% of drug candidates in the drug development process and 40% of the top 200 oral drugs in the US and European markets are facing solubility challenges [2]. To alleviate the burden of drug discovery and development and improve the clinical efficacy, more effective approaches are in urgent need of overcoming the solubility issue of poorly soluble drugs.

Amorphous solid dispersions (ASDs), are a solid-state dispersion with an amorphous active pharmaceutical ingredient (API) molecularly dispersed in pharmacologically inert carriers. ASDs are regarded as one of the most promising and efficient techniques for the solubilization of poorly water soluble APIs [3,4]. ASDs have been widely studied, and there have been a huge number of high quality reviews summarizing the development and research progress of ASDs [5,6,7,8,9,10,11]. Thus, this review only provides a very brief introduction to ASDs, which leads to the main focus of the review, i.e., microwave induced in situ amorphization, as an upcoming approach to solving the physical stability concerns of ASDs.

As predicted by Hancock et al., amorphous APIs exhibited 10 to 1600 times higher solubility than their most stable crystalline counterparts due to their higher Gibbs free energy [12]. However, such difference in free energy is also regarded as a thermodynamic driving force for recrystallization [13]. ASDs can benefit from the solubility advantage of amorphous APIs and meanwhile enhance the stability of amorphous drugs with the assistance of polymeric carriers, where polymers act as anti-plasticizers and reduce the collisions between drug molecules and help with the supersaturation maintenance by virtue of their molecular coupling with the dissolved API, steric hindrance and higher viscosity in solution [14,15].

Despite the extensive in-depth studies carried out in ASDs, commercialization of ASDs is still very limited (approximately 30 marketed products since 1975) [9,16,17,18]. This situation may stem from a range of problematic issues of ASDs, mainly including physical stability, downstream processing and scale-up difficulties [19,20,21]. In 2017, microwave-induced in situ amorphization was firstly proposed by Doreth et al. as a promising approach to tackling these problems [22]. This technique can be utilized to circumvent the physical stability issue and simplify the problematic downstream processing of ASDs by preparing a final dosage form containing the crystalline drug and subsequently “activating”, i.e., amorphizing, the drug on demand within its final dosage form prior to administration using household microwave ovens under an optimized and well defined operation setting, including microwave power and time. The aim of this review is to provide a systemic overview of the state-of-the-art of microwave-induced in situ amorphization as a novel ASD preparation strategy as well as highlighting current challenges of this technology.

## 2. Current Issues in the Development of ASDs

### 2.1. Stability Issues of ASDs

Among all the challenges faced by ASDs, the issue of physical stability during manufacture and storage has attracted widespread attention, since recrystallization of the amorphous drug from the ASDs will result in a considerable loss of the solubilizing effects of the ASDs. The main factors affecting the physical stability of ASDs are discussed in the following sections.

#### 2.1.1. Thermodynamic Factors on the Physical Stability of ASDs

The thermodynamic driving force for recrystallization of amorphous APIs derives from a tendency of the amorphous system with high free energy and excess enthalpy and entropy to relax towards a lower free energy [23]. It has been suggested that when the amorphous API and polymer are completely miscible and the drug loading is equal to or below the saturation solubility of the crystalline API in the polymer, the ASDs are physically stable with no thermodynamic driving force for crystallization [24,25]. However, such a drug loading is often too low to satisfy the pharmaceutical dose required [26]. When the drug loading exceeds the drug–polymer solubility limit, the system becomes thermodynamically unstable, which carries the risk of amorphous–amorphous phase separation and/or drug recrystallization during manufacture and storage of such ASD systems. Therefore, from the thermodynamic perspective, polymers that offer high drug–polymer solubility are favored as carrier candidates for generating stable ASDs. Some findings suggested that a strong drug–polymer interaction could lead to a reduction in the Gibbs free energy and enthalpy of mixing, which increased the drug–polymer miscibility/solubility [27]. Such intermolecular interaction was also reported to be a critical attribute to the increase of configurational entropy of API molecules, which promoted the thermodynamic stability of ASDs by connecting the drug and the polymer as a single entity [28].

#### 2.1.2. Kinetic Factors on the Physical Stability of ASDs

From the kinetic point of view, the molecular mobility of amorphous materials mainly results from α-relaxation [24], i.e., the glass transition temperature (*T_g_*), which is related to the crystallization time and the crystallization rate of amorphous materials, respectively [25,26].

A key factor contributing to the physical stability of ASDs is their molecular mobility, which is mainly represented by the primary (α-) relaxation [29], i.e., the general mobility change responsible for the *T_g_*. In order to slow down the molecular mobility of amorphous materials, a rule of thumb is that the storage temperature of such materials should be 50 K lower than their *T_g_* (*T_g_* – 50 rule). Since drugs usually show a low *T_g_*, a large quantity of high *T_g_* polymers needs to be included in the formulation to considerably increase the *T_g_* of the resultant ASD systems (known as anti-plasticizing effect) for kinetic stability enhancement [8]. Again, the high polymer ratio in the formulation may result in a low drug loading. Additionally, polymers with a high *T_g_* may cause difficulties in processing. For instance, the use of such high *T_g_* polymers in hot melt extrusion (HME), one of the most popular preparation methods for ASDs, usually requires the aid of plasticizers, which in turn will reduce the systemic *T_g_* [30,31]. Polymers that interact with APIs through hydrogen bonds and ionic interactions have been shown to restrict the molecular mobility of ASD systems in addition to their contribution in thermodynamic stability [11,32].

#### 2.1.3. Environmental Factors Affecting the Physical Stability of ASDs

Environmental factors including temperature, humidity and mechanical stress are indirect factors affecting ASDs stability thermodynamically and/or kinetically. As reported by Lin et al., when the environmental temperature was elevated, there could be orders of magnitude increases in molecular mobility according to the Adam–Gibbs–Vogel equation [8], resulting in an accelerated phase separation and recrystallization of ASDs. Regarding the ambient humidity, it can introduce a certain amount of moisture to the hygroscopic ASD systems. Kinetically, moisture can lead to a plasticizing effect (reducing the systemic *T_g_*), thereby increasing overall molecular mobility and the risk for recrystallization. Furthermore, the absorbed water has the potential to disrupt drug–polymer interactions by competing with hydrophilic polymers to form hydrogen bonds with APIs [33]. It was also implied in some studies that moisture uptake could reduce the drug–polymer solubility, and the extent of such effects varied with different APIs [34,35].

ASDs can also be sensitive to mechanical stress, such as grinding, crushing, scratching or compressing during the formulation process, which can facilitate the deformation-induced molecular mobility in amorphous APIs and result in phase separation or crystallization [11,36]. For instance, milling of ASD extrudates after HME was observed to destabilize supersaturated ASD systems compared to ASDs below the saturation solubility in the polymer [37].

### 2.2. Downstream Processing Issues of ASDs

Downstream processing of ASDs, typically consisting of feeding, blending and tableting, can have an impact on the physical stability of ASDs by influencing the environmental factors, as described in Section 2.1.3, as well as the content uniformity of final products, as reviewed in detail by various research groups [6,38]. The latter issue with content uniformity is mainly ascribed to the poor flow properties of bulk ASD powders for further processing. In this regard, current studies tend to focus on alternative manufacturing strategies such as electrospinning, HME-injection molding and an HME–3D printing combination, but the application of such approaches is often limited by the poor scalability as compared to traditional formulation lines [6,39].

## 3. In situ Amorphization

### 3.1. The Pharmaceutical Significance of in situ Amorphization

With regard to the aforementioned difficulties in ASD development, it is worth highlighting that, although various solutions were proposed to promote the stability of ASDs, conventional ASD systems with low drug–polymer solubility (usually lower than the drug loading) may always face potential phase separation and recrystallization risks during manufacture and storage [26]. Even for thermodynamically stable ASD systems, environmental factors such as high humidity are likely to affect the drug–polymer miscibility and hence decrease the drug solubility in the polymer. However, if the APIs can exist in their stable crystalline form within their formulations during the manufacturing process and storage and subsequently be amorphized “in situ” on demand prior to administration, the physical stability of ASDs is in theory only required during a very short period from in situ amorphization to administration. Hence, the application of the in situ amorphization approach can not only circumvent the physical stability concerns of ASDs but also avoid additional intermediate ASD preparation steps such as spray drying or HME as well as issues related to the downstream processing of ASDs, particularly the poor flowability [40,41].

### 3.2. General Development of in situ Amorphization

As reviewed by Priemel et al. in 2016 [26], in situ amorphization was initially depicted as an "unintended" spontaneous amorphization event taking place prior to or after the administration of poorly soluble crystalline drugs without deliberate external energy input (see Table 1 for a summary of the latest literature supplemented). Here, in situ amorphization occurred spontaneously due to the exposure of formulations to certain inducements and could be divided into three main amorphization pathways: (i) dissolution-mediated approaches (dissolution of the crystalline drug into a polymeric matrix in the presence of moisture or methanol vapor); (ii) vaporization-mediated approaches (adsorption of sublimated drug onto porous excipients); and (iii) in situ amorphization during lipolysis (mimicking in vivo digestion) of lipid-based oral formulations. Among the three pathways, dissolution-mediated in situ amorphization utilizing tablets as the final dosage form is believed to be the most promising application for "intended" in situ amorphization, since it directly applies a final dosage form (i.e., tablets) rather than intermediate products such as physically mixed powders or the lipid-baseddrug delivery systems that have the potential issue of unintended partial amorphous precipitation [42].

Despite the application prospects of the above mentioned in situ amorphization approaches, other issues could also arise, such as insufficient degree of amorphization, drug degradation, insufficient disintegration and, most importantly, the long timeframes required to obtain the desired in situ amorphization effects with such approaches, which varied from hours to months (Table 1). In order to accelerate the pace and improve the controllability of in situ amorphization, microwave irradiation, a fast and easy-to-use technology, has recently caught attention [43,44,45]. Some initial attempts have been made using microwaves to induce ASDs in situ on the basis of dissolution-mediated in situ amorphization [22,46,47]. Hence, in the following sections, microwave-induced in situ amorphization is reviewed in more detail.

## 4. Microwave-Induced in situ Amorphization

### 4.1. Microwave Heating

Microwaves are electromagnetic waves with high frequencies (0.3–300 GHz) and wavelengths in the cm range, i.e., 1–0.001 m, which have been widely used in the fields of communication, radar, industry, scientific research, medicine and domestic life [52]. Microwave heating systems are usually restricted within the industrial, scientific and medical (ISM) frequency bands, using certain license-free frequencies designated by the International Telecommunication Unit (ITU) with international agreements [53]. Among them, the most commonly used frequency for commercial and domestic uses in most countries is 2.45 GHz with a wavelength of 0.122 m and a penetration depth of 0.014 m in water at 25 °C [54,55].

In the pharmaceutical industry, microwave heating is mainly applied to modify the physicochemical characteristics and drug delivery performances of dosage forms (e.g., SDs, orally disintegrating tablets, film coating preparations and gel beads) through microwave-assisted drying, curing, fusion, sintering, crosslinking etc. [56,57]. It is well known that in conventional heat treatments, energy is transferred via heat convection, conduction and radiation from the surface to the body of a material. In contrast, in microwave heating, energy is transferred directly to both the surface and the body of a material through molecular interactions in the electromagnetic field, with energy converted directly from the electromagnetic energy into heat, which can be primarily explained through the dipolar polarization mechanism [58]. To be able to absorb microwaves, a material needs to have suitable dielectric properties, i.e., a molecule that exhibits a sufficiently strong dipole moment and is sufficiently small or flexible to rotate within the material structure, such as water [59]. Heating is achieved by coupling microwave radiation to the dipole moment of the molecule and exciting (hindered) rotational motions of these molecules within the material [60]. Specifically, when a direct current electric field is applied, the dipoles of the material will change from a disordered and irregular state to a state with specific orientation and generate potential energy. If the direction of the electric field is changed rapidly and alternately (with alternating current applied), dipoles will oscillate rapidly in order to track the electric field. The ease of the movement of dipoles in the field depends on the viscosity and the mobility of the electron clouds. Due to the molecular thermal motions and the intermolecular interactions, the regular oscillation of dipoles can be disturbed or hindered, resulting in a friction-like effect that enables the molecules to obtain energy and release it in the form of heat. Consequently, the temperature of the material increases, which is known as the specific effects of microwave [61]. The above mechanism endows microwave heating distinct characteristics from conventional heating, such as rapid volumetric heating, selective heating (high dielectric substances are favored), low energy consumption, easy operation and low preparation costs [44,62]. The conversion of the microwave energy to heat can be described with the following formula (Equation (1)) according to the electromagnetic field theory:(1)$$P=2\pi f\varepsilon_{0}\varepsilon_{r}^{’’}E^{2}$$
where *P* is the microwave power dissipation (W∙m^–3^); *f* is the frequency and *E* is the intensity of the electric field; *ε_0_* is the permittivity of free space (a vacuum), whose value is 8.854 × 10^−12^ F∙m^−1^; *ε_r_″* is the imaginary part of the complex dielectric permittivity, which refers to the dielectric loss and represents a major parameter related to the microwave heating [63].

The formula suggests that a high frequency and a strong intensity of the applied electric field can contribute to more heat generation (high *P* value), and materials with high dielectric loss can be more efficiently heated up in the microwave field.

### 4.2. Microwave Heating for Bulk Preparation of Amorphous Glass Solutions

It is not surprising that microwaves have also been introduced for the preparation of ASDs. However, unlike water, polymers and drug molecules are poor microwave absorbers [64,65] and hence do not heat up sufficiently on their own in order to form an ASD. As a consequence the preparative methods described in the literature typically take advantage of the following effects: i) convective heating, i.e., a microwave absorbing reactor or sample holder is heated up and this heat is then transferred to the sample indirectly [43,44,66,67,68,69,70,71,72,73,74,75], and ii) addition of a microwave absorbing solvent to the formulation in excess and subsequent solvent evaporation from a drug–polymer solution [76] or solvent slurry [77,78]. In both cases, microwave heating is merely replacing traditional heating methods and hence, adds little specific benefits compared to preparative methods such as hot melt extrusion and spray drying. In addition, the prepared ASDs require further processing and handling into a pharmaceutical dosage form, such as a tablet, and most importantly they face the same stability issues as conventionally prepared ASDs upon storage.

Microwave heating has been utilized as an alternative fusion-based (convective heating) ASD preparation method since 2008, when it was used to induce amorphization of IBU in the hydrophilic carriers including polyvinylpyrrolidone/vinyl acetate 64 (PVP/VA 64) and HP-β-cyclodextrin (β-CD) [44]. The authors found that physically mixed IBU:PVP/VA 64 1:1 (*w/w*) powders could turn partially amorphous when microwaved for 6 min at 600 W. A similar system composed of IBU:β-CD at a 1:2 molar ratio reached 100% amorphicity after microwaving at 600 W for 15 min. In dissolution tests, microwave processed powder systems with high amorphicity incorporating either carrier showed enhanced dissolution behavior in water compared to IBU, IBU-carrier physical mixtures and mixtures of separately microwaved IBU and carriers. Since then, several studies showed the viability of using microwaves to prepare bulk ASD powders, as shown in Table 2. In general, the primary process of the microwave-induced amorphization involves three steps:(i)Preparing the physically mixed API-carrier powders. It is worth noting that low *T_g_* carriers (polymers, surfactants or the combination of both) have frequently been used (Table 2) as they can possibly be molten or softened at relatively low temperatures;(ii)Treating the physically mixed loose powders with continuous or intermittent microwaves to induce amorphization; and(iii)Cooling, pulverizing and sieving to get the final bulk ASD powders.

With the application of microwave heating in ASDs, an evident advantage appears over most of the other fusion-based methods, where relatively long exposure under high temperature is required to melt the drug and carrier, which may lead to degradation of some thermally unstable components [79]. In the case of microwave heating, the processing time is usually less than 15 min, greatly reducing the exposure time and potentially protecting the components from degradation [80,81].

Most of the studies summarized in Table 2 used microwave absorbing reactors or sample holders, i.e., materials that are heated during microwave processing, such as beakers and crucibles, which in turn heat up the sample through convective heating. Hence, it is likely that the amorphization of the drug follows the same conductive heating mechanism as that of the conventional fusion-based ASD preparation methods. This means that the effect of conductive heating on drug amorphization cannot be completely excluded when analyzing the mechanisms of microwave-induced amorphization, and the aforementioned benefits of this technology may not be fully ascribed to microwave heating within the sample itself. Therefore, the use of microwave transparent sample holders, such as polypropylene (PP) and polytetrafluoroethylene (PTFE) [22,64,82], will help to better understand the role of microwave irradiation in microwave-induced ASD preparation.

Microwave heating has also been applied to process drug–polymer solutions and slurries using high-loss polar solvents for the fabrication of bulk ASD powders. For example, water has been used in API–hypromellose acetate succinate (HPMCAS) and API–HPMCAS–urea physical mixtures to prepare homogeneous slurries, which heated up and evaporated during microwaving to induce full amorphization of the API [77,78]. In another study, Radacsi et al. prepared a niflumic acid–PVP–ethanol solution, which was subjected to a well-designed microwave-assisted evaporation setup with a constant power of 300 W [76], and a completely amorphous ASD was formed, showing a significantly enhanced dissolution performance relative to crystalline niflumic acid.

### 4.3. Microwave-Induced in Situ Amorphization within the Final Dosage Form

As outlined above, microwave heating was demonstrated to be a feasible alternative in the manufacture of bulk ASDs. However, like traditional ASD preparation methods, fabrication of intermediate loose ASD powders using microwave heating still cannot overcome the physical stability and downstream processing issues of ASDs. The true potential of using microwaves to induce amorphization lies in its applicability on the final dosage form. In other words, if microwave-induced amorphization can occur within the final dosage form, i.e., in situ, the consistent physical stability and downstream processing issues of ASDs can be avoided while amorphous drugs are formed to present enhanced dissolution performances. This novel concept was introduced by Doreth et al. in 2017 as an in situ amorphization process where a given drug is amorphized via microwave irradiation within its final dosage form [22]. Since it is still a new approach, there are currently only few publications available on microwave-induced in situ amorphization (all using PP as the microwave transparent sample holder), which are summarized in the following four subsections.

#### 4.3.1. The Role of Microwave Energy Input and Storage Humidity

In the first proof of concept study by Doreth et al. in 2017, a significant impact of microwave energy input and storage humidity of final dosage forms on the degree of microwave-induced in situ amorphization was found in the IND:PVP systems (Figure 1) [22]. In this study, compacts containing a powder mixture of crystalline IND, the polymer PVP K12 (drug:polymer 1:2 (*w/w*)) along with a glidant and a lubricant (colloidal silicon dioxide and magnesium stearate, respectively) were prepared and subjected to microwave irradiation. It was initially found that plain compacts without further treatment did not amorphize due to the poor electro-chemical coupling of the drug and polymer, and hence, the poor response when subjected to the microwaves. Because of this, the authors introduced different amounts of absorbed water by conditioning the compacts at different RH (32, 43 and 54% RH) for at least two weeks. The absorbed water then acted as the microwave absorbing entity in the compacts. The study also made use of different microwave energy inputs based on various combinations of the irradiation power (300, 600 and 1000 W) and the processing times (50, 90, 150 and 300 s). It was observed that the degree of amorphization of IND increased as the amount of absorbed water in the compacts and microwave energy increased. The highest IND amorphicity around 80% was achieved with compacts containing the highest amount of water (9.8% (*w/w*)) while applying the highest energy input (90 kJ). The authors suggested that a complete amorphization was not achieved upon microwaving, because the absorbed moisture had evaporated too fast from the surface of the compacts, resulting in a surface with low water content, which did not sufficiently absorb the microwave energy anymore for the amorphization process. Hence, a shell layer containing the crystalline drug was formed, whereas the core of the compacts could be amorphized during microwaving. Overall, the study showed that in situ amorphization using microwaves can be achieved in the presence of a microwave absorbing entity in the compacts, i.e., absorbed water.

#### 4.3.2. The Role of the Molecular Weight of Polymeric Carriers

In a follow-up study by Doreth et al. (2018), the authors investigated the impact of the molecular weight of the polymer PVP on the degree of amorphization upon microwaving [46]. Specifically, PVP K12, K17 and K25 with different molecular weights (2000–3000 g∙mol^−1^, 7000–11,000 g∙mol^−1^ and 28,000–34,000 g∙mol^−1^, respectively) were studied as the model polymeric carriers. Compacts containing the crystalline drug and the different PVP grades at a ratio of 1:2 (*w/w*) were initially conditioned at 54% RH and subsequently subjected to the microwave energies of 90 kJ and 180 kJ. It was found that PVPs with different molecular weights exhibited similar hygroscopicity reflected by the similar moisture uptake (approx. 10% (*w/w*)) in all compacts after storage. However, the morphology of the compacts was different when using different PVP grades. As illustrated in Figure 2, the compacts containing PVP K12 presented the greatest reduction in porosity after the microwaving treatment as compared to those containing PVP K17 and PVP K25. It was suggested that this morphological change might have led to less moisture evaporation, which explained the lowest moisture loss after microwaving observed with IND:PVP K12 compacts among the three formulations. Hence, the highest drug amorphicity was found in IND:PVP K12 compacts, namely 31% with 90 kJ energy input and 58% with 180 kJ energy input, whereas compacts containing PVP K17 and K25 showed almost no crystallinity decrease after exposure to microwave irradiation with either energy inputs. Moreover, the low porosity of the microwaved IND:PVP K12 compacts can be associated with the initial delayed release of such compacts in the dissolution assessment, which is further discussed in Section 4.3.4.

It was also suggested that the superiority of PVP K12 in promoting amorphicity could be explained by the fact that PVP K12 was the only polymer in this study possessing a *T_g_* below RT after storage due to the plasticizing effect of water, implying the transformation from glassy to supercooled liquid state occurred with an increased chain flexibility. The enhanced chain flexibility then led to an increase in the molecular mobility of water molecules within the conditioned IND:PVP K12 compacts, which facilitated the molecular friction and heat generation upon microwaving. Moreover, further investigation demonstrated that such a plasticizing effect of water was advantageous not only in microwave-induced in situ amorphization, but also when compacts were exposed to convective heating [85].

#### 4.3.3. Dissolution of the Drug into the Polymeric Carrier

In a recent study by Edinger et al. (2018) the success of microwave-induced in situ amorphization was suggested to be related with the dissolution of the drug into the polymer upon microwaving, where the polymer acted as the "solvent" for the crystalline drug [47]. Here, compacts containing physically mixed celecoxib (CCX), PVP K25 at drug loadings of 10, 20, 30, 40 and 50% (*w/w*) together with 0.5% (*w/w*) magnesium stearate were compressed, stored at RT, 75% RH for 4 weeks and subjected to microwave irradiation at 1000 W in 60 s intervals for 10 min. Microwave-induced in situ amorphization was mostly observed to take place in the first 4–5 min of microwave processing, and higher degrees of amorphization were obtained for lower drug loadings. Nevertheless, full amorphization could not be achieved even at 10% (*w/w*) drug loading. Since the drug needs to dissolve into the polymer network during microwaving, the assumption proposed in this study was that the lack of full amorphization might have been due to the large particle size of the drug crystals, which resulted in the formation of a saturated drug–polymer diffusion layer that hindered the further dissolution of the drug into the polymer. In addition, the likelihood of forming drug particle clusters increases with the increase in drug loading, leading to insufficient direct contact between the drug and polymer particles and further hindering the amorphization process.

Therefore, in a subsequent study by Hempel et al. in 2020, the particle size of drug and polymer was reduced to improve drug–polymer contact and enhance the dissolution of the crystalline drug into the polymer [86]. Using this approach, it was possible for the first time to obtain a full amorphization of the drug CCX. The starting materials, CCX and PVP (Kollidon^®^ 12PF), were prepared in two particle size fractions, either small (<71 µm, ball-milled and sieved) or large (>71 µm, sieved), and compacts containing CCX:PVP:magnesium stearate at a 30:69.5:0.5 ratio (*w/w*) were stored under the same condition as reported by Edinger et al. [47]. Microwave parameters were set at 1000 W in 60 s intervals for 10 min or continuously for 10 min. It is noteworthy that the 100% amorphicity was achieved only in tablets with small CCX and PVP particles subjected to continuous exposure of microwaves, which was 76% higher than the amorphicity of the same tablets under interval exposure and approx. 30–75% higher than that of tablets with other particle size combinations under continuous microwave exposure. The patterns of microwave exposure, interval or continuous, was suggested to have an influence on the total exposure time of the compacts above the *T_g_* of the polymer and the emerging ASD. Continuous microwaving resulted in a longer total effective exposure time above the *T_g_* of the system and, hence, led to an overall longer time for the crystalline drug to dissolve into the polymer. On the basis of the above findings, the authors proposed that both the particle size of components and effective microwave exposure time contributed to the amorphization by improving the dissolution of the drug in the polymer. However, the former had a more significant influence on microwave-induced in situ amorphization than the latter, which was demonstrated by the fact that compacts consisting of components with large particle sizes could only achieve half the amorphicity of the small particle size group despite an additional 300 s of effective microwave exposure (illustrated in Figure 3).

The importance of the dissolution of drug in polymer was also addressed by Hempel et al. in a comparative study of microwave heating and convective heating to induce in situ amorphization [85]. Two experimental setups were applied in the study, including a microwave-induced method identical to the one that successfully led to 100% CCX amorphicity, as previously described [86], and a comparable convection-induced method that imitated the setup of the former. Results showed that for conditioned (RT and 75% RH for 2 weeks) CCX:PVP tablets, the drug amorphicity in the convection heating group failed to reach 100% relative to the full ASD obtained in the microwave heating group even when applying 3-fold longer heating times. This was primarily due to the fact that the convection heating group exhibited a delayed temperature profile as compared to the microwave heating group arising from their different heating mechanisms, which resulted in a delayed dissolution process of CCX into the PVP and ultimately an incomplete in situ amorphization.

#### 4.3.4. Performance of Microwave-Induced ASDs Activated in situ

Doreth et al., in 2017, found that no microwave-induced drug degradation was detected in the IND:PVP system after all processing treatments [22]. Furthermore, it was found that the intrinsic dissolution rate of the microwaved tablets with the highest amorphicity (80%) was 6.3-fold higher than that of the physical mixture containing crystalline IND and PVP. More importantly, even though no full amorphization was achieved, the dissolution rate was comparable to that of the quench-cooled fully amorphous samples. In another publication in 2018, Doreth et al. evaluated the in vitro dissolution of microwaved compacts under non-sink conditions at pH 6.8 [46]. As shown in Figure 4, microwaved IND:PVP K12 compacts exhibited a slower initial release of IND compared to other formulations, which was possibly related to the low porosity and high hardness as well as a lack of disintegration that took approx. 4 h in comparison to the high porosity, low hardness and approx. 10 min disintegration time of microwaved IND:PVP K17 and IND:PVP K25 compacts. Such a phenomenon has been previously described in relation to the "soft and waxy nature" of ASDs by Serajuddin et al. [87], where polymeric carriers are known to be strong binders within tablets and can reduce the porosity of tablets when they are softened, molten or plasticized during compression. Nevertheless, microwaved IND:PVP K12 compacts showed the highest extent of drug release among all formulations studied after 3 h mainly due to the highest IND amorphicity, as previously stated in Section 4.3.2. To alleviate this issue, disintegrants or porogens potentially need to be added to the formulation in future investigations.

## 5. Challenges and Future Perspectives

Microwave-induced in situ amorphization is a very promising and novel method to prepare ASDs in final formulations. However, to date there have been only a limited number of investigations addressing the pharmaceutical applications of such a new technology where many facets still require further investigations, including the underlying mechanisms and influencing factors of microwave-induced in situ amorphization (the main focus of current publications); the chemical stability of materials prior to, during and post processing; the selection criteria of excipients; the practicality of the dosage form after processing; the reproducibility of the microwaved formulations and a feasible control unit/indicator that allows the pharmacist and patient to identify whether a successful amorphization has been achieved after processing; as well as the clinical application of this technology.

With respect to the chemical stability of the drug, there is a risk of drug degradation during microwave processing due to the applied microwave energy on the sample. However, even prior to microwaving, the presence of absorbed water, which is a prerequisite to enable in situ amorphization by microwave irradiation, can potentially lead to hydrolysis and drug degradation. It should be noted that the reason why no degradation of IND was detected in the study by Doreth et al. might be due to the short storage period (RT and 54% RH for two weeks) prior to microwaving at 1000 W for 90 s [22]. Considering the large amount of absorbed water (approx. 10%) present in such systems, it is likely that long term storage may have a significant impact on the chemical stability profile. Therefore, more systematic investigations into the drug degradation during storage and microwave processing need to be conducted.

Up to date, only a limited number of polymers have been explored in this field with a main focus on PVP K12. In order to study a broader applicability of the concept, it is critical to study other polymeric and non-polymeric carriers. In this context, the type of carrier material may have an impact on the hardness and disintegration properties of tablets in addition to the drug amorphicity upon microwaving. For example, even though fully amorphous systems could be obtained with PVP K12, there are still some challenges with respect to the dosage form, its preparation and processing: (i) form stability issues of the compacts investigated so far, which deformed primarily due to water evaporation and became highly dense with low porosity after microwaving [22,46,47]; (ii) the tablet size and compaction pressure during production presumably have impacts on the amorphization kinetics; (iii) APIs with different glass forming abilities might show different performance in microwave-induced in situ amorphization; and (iv) the influence of the precise position of the tablets in microwave ovens on microwave-induced in situ amorphization. All these issues require more in-depth studies in the future.

Regarding a potential future clinical application of this technology, it is critical that the quality of microwave-able dosage forms is ensured by health care professionals, such as pharmacists, before distributing the medicine to the patients. On one hand, the manufacturing and packaging of the products should be carefully designed (e.g., using moisture substitutes or encapsulation) for microwave processing in order to avoid changes in the optical appearance of the dosage forms whilst providing full amorphization with the desired dissolution performance. On the other hand, a standard operation procedure, including the microwave setup and processing, should be developed to guarantee good reproducibility of the microwaved formulations. In addition to in vitro dissolution studies that were investigated in the previous studies, in vivo drug release and pharmacokinetic properties of the microwave-induced ASDs should also be investigated with a final view to further translating such a promising technology.

## 6. Conclusions

This review has provided an overview of the current stage and future perspectives of microwave-induced in situ amorphization. In situ amorphization has previously been observed, mostly unintentionally, during manufacturing, storage, dissolution and lipolysis; however, when using microwave irradiation, one can use it intentionally and on demand. Even though the generation of microwave-induced ASDs is still a fairly new technology with current challenges, it has significant potential of providing a novel means of preparing ASDs and at the same time circumventing stability or downstreaming issues associated with the current manufacturing routine for ASDs.

## Figures and Tables

**Figure 1 pharmaceutics-12-00655-f001:**
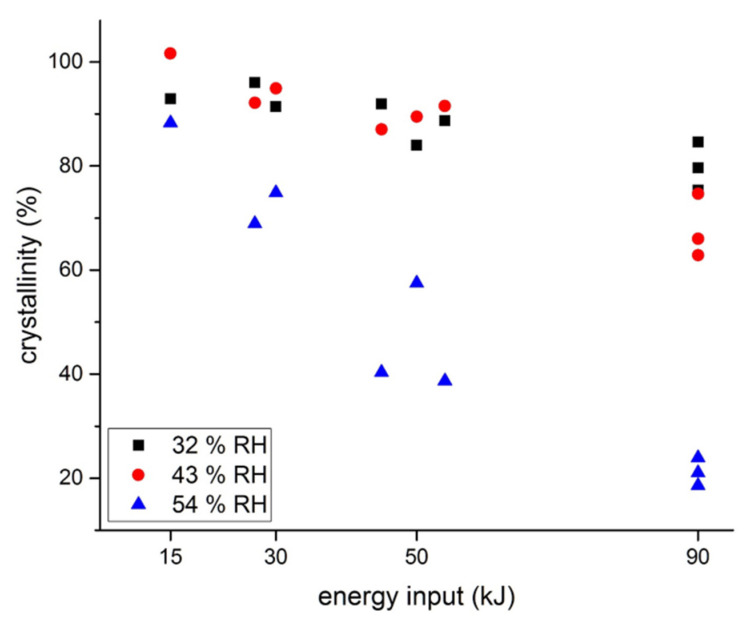
Crystallinity of IND (%, *w/w*) versus energy input during microwaving of compacts that had been stored at 32%, 43% or 54% RH and microwaved at nine different settings. (Reproduced from [22]).

**Figure 2 pharmaceutics-12-00655-f002:**
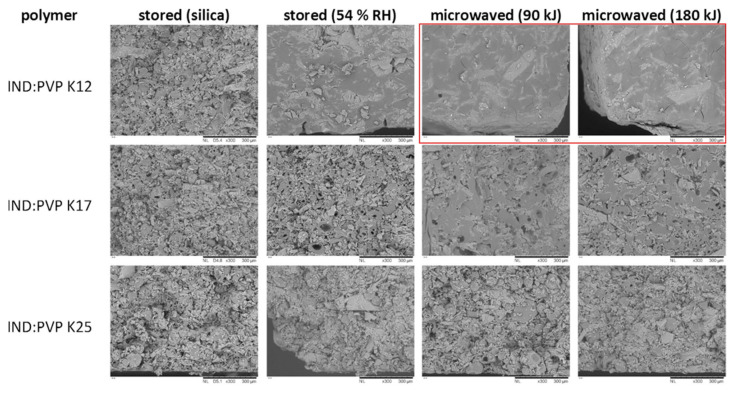
Scanning electron micrographs of the cross section of IND:PVP K12, IND:PVP K17 and IND:PVP K25 compacts after storage under dry conditions, 54% RH and after subsequent processing with 90 kJ and 180 kJ microwave energy. (Reproduced from [46]). The highest extent of pore network reduction is highlighted by the red square.

**Figure 3 pharmaceutics-12-00655-f003:**
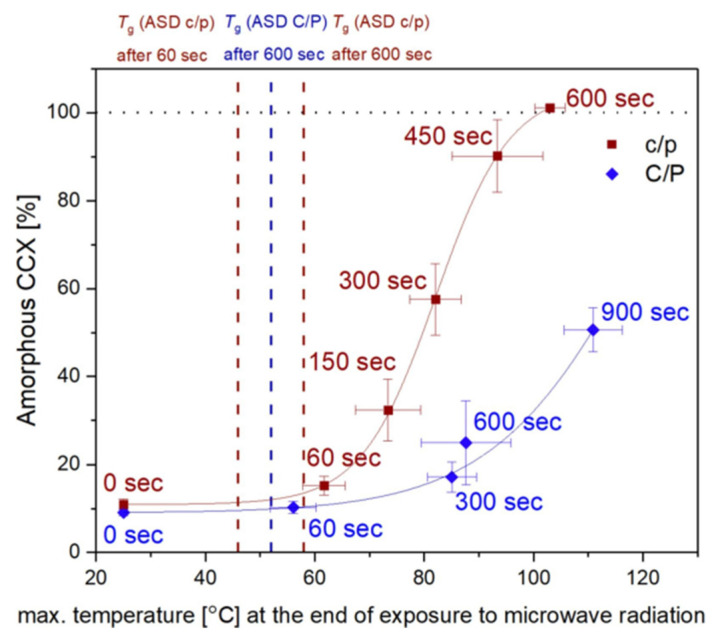
Degree of amorphization for the drug CCX plotted against the compact temperature obtained during microwaving at different exposure times. Two compacts containing CCX and the polymer PVP K12 with different particle sizes are shown, where c/p and C/P refer to small and large sizes of CCX/PVP, respectively. Mean ± SD (*n* = 3). The dashed lines indicate different *T_g_*s. (Reproduced from [86]).

**Figure 4 pharmaceutics-12-00655-f004:**
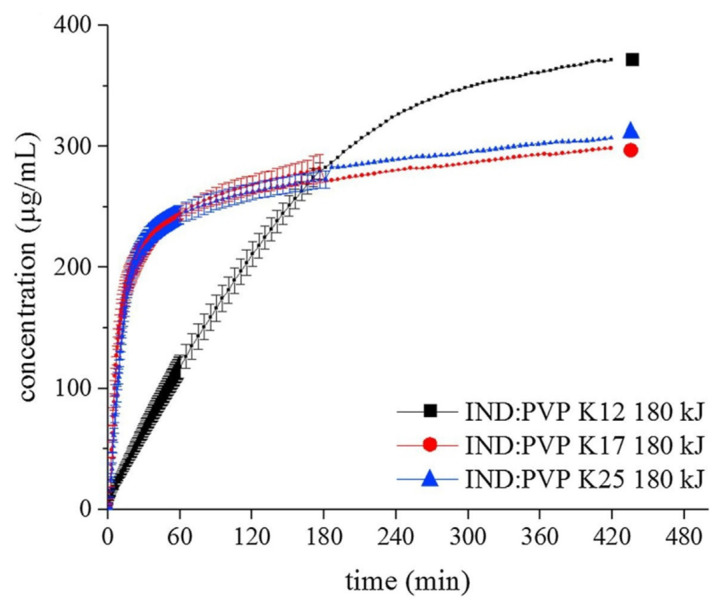
Dissolution profiles of IND:PVP K12, IND:PVP K17 and IND:PVP K25 compacts after microwaving with an energy input of 180 kJ. (Reproduced from [46]).

**Table 1 pharmaceutics-12-00655-t001:** The applications of non-microwave induced in situ amorphization in oral drug delivery.

Category	API	Upstream Processes	Inducements of in situ Amorphization	Post-Amorphization Dosage Form	Impact of in situ Amorphization	Ref.
Dissolution-mediated in situ amorphization	Paracetamol, ibuprofen (IBU), ketoprofen and naproxen	Mixing (API–polyethylene oxide), compaction	Moisture uptake during storage (room temperature (RT), 94% relative humidity (RH)) of compacts for 3–4 weeks	Compact	NA	[26]
	IBU	Dry mixing for 3 h (API–hydroxypropyl methyl cellulose)	Methanol or water spray followed by milling for 1–3 h; storage in saturated methanol vapor for 16 h; saturated methanol vapor followed by milling for 16 h	Powder	NA	[26]
	Indomethacin (IND)	Dry mixing for 3 h (API–hydroxypropyl methyl cellulose)	Saturated methanol vapor followed by milling for 16 h			
	IND	Mixing (API–Eudragit^®^ E), compaction	Immersion in pH 6.8 phosphate buffer for 1–3 h at 37 °C	Compact	Improved dissolution behavior when the swelling degree was high	[26]
	Naproxen, IBU	Mixing (API–Eudragit^®^ E), compaction	Immersion in 50 mL purified water for 1 h	Compact	Improved dissolution behavior	[48]
	IND	Mixing (API–L-arginine), blending, compaction, coating (Kollicoat^®^ Protect)	Moisture uptake during storage (RT, 75% RH), not completely amorphized even after 91 days	Coated compact	Good mechanical stability; improved bioavailability	[40]
	IND	Mixing (API–L-arginine)	Moisture uptake during storage (RT, 75% RH), not completely amorphized even after 101 days	Powder	Chemical degradation	[49]
	Furosemide, IND	Mixing (API–L-arginine), blending, compaction, coating (Eudragit^®^ L)	Immersion in 0.1 M HCl for 10–30 min at 37 °C (co-amorphization); recrystallization after longer immersion times	Coated compact	L-arginine induced chemical degradation of IND (prevented by adding citric acid but at the expense of amorphization)	[50]
	Carvedilol	Mixing (API–L-aspartic acid), blending, compaction, coating (Eudragit^®^ L)	Immersion in 0.1 M HCl for 45 min at 37 °C (co-amorphization)	Coated compact	Insufficient disintegration and poor drug release	[51]
Vapor-mediated in situ amorphization	Aspirin, phenacetin	Mixing (API–magnesium aluminum silicate/activate carbon)	Storage at 25 °C with silica gel for 1–2 week; storage at 40 °C with reduced pressure for 2–8 h	Powder	NA	[26]
	IBU	Mixing (API–silica)	Storage at 40 °C, 0% RH for 4–5 weeks; hydrophilic, small pore diameter silica was preferred	Powder	Potential instabilities due to drug molecules migration and subsequent interaction with excipients	[26]
	Diflunisal	Mixing (API–silica)	Storage at 80 °C, 0% RH for 2–3 weeks			
In situ amorphization during lipolysis	Cinnarizine, halofantrine and simvastatin	Preparing the self-microemulsifying drug delivery systems (SMEDDS) or the self-nanoemulsifying drug delivery systems (SNEDDS)	Immersion in lipolysis medium for approx. 1 h	SMEDDS or SNEDDS	Unintended partial amorphous precipitation. Lack of correlation of in vitro precipitation to the in vivo performance	[26]

NA: not available.

**Table 2 pharmaceutics-12-00655-t002:** Summary of publications on the application of microwave heating in the preparation of bulk amorphous solid dispersion (ASD) powders.

Year	Microwave Instrument	Power (W)	Processing Time (min)	Microwave Absorbing Sample Holder	API	Carrier	Cooling Method	Drug Released %	Ref.
2008	CE297DN, Samsung	600	6	Beaker	IBU	β-CD; PVP/VA 64 (did not achieve fully amorphous)	NA	90 (2 min)	[44]
2009	CE297DN, Samsung	600	15; 9	Beaker	Nimesulide	Gelucire^®^ 50/13; Poloxamer 188	NA	>90 (16 min); >90 (70 s)	[66]
2010	CATA-2R, Catalyst Systems, Pune, India	590	3; 4; 5; 6	Beaker	Atorvastatin calcium	PEG6000	RT	52; 57; 61; 64 (120 min)	[67]
2010	CE297DN-Samsung, Surrey, England	600	10	Beaker	Itraconazole	D-α-tocopheryl polyethylene glycol 1000 succinate	RT	>90 (2 min)	[68]
2011	Modified domestic microwave oven	Variable power	Around 30	Crucible	IBU	Stearic acid; PVP 40	RT	>60 (20 min)	[43]
2012	CATA-2R, Catalyst Systems, Pune, India	600	3; 4; 5; 6	Glass beaker	Repaglinide	Poloxamer 188	NA	68; 73; 80; 82 (60 min)	[69]
2013	CATA-2R, Catalyst Systems, Pune, India	590	3; 4; 5; 6	Beaker	Raloxifene	HPMC E5 LV	RT	50; 57; 60; 65 (120 min)	[70]
2013	CATA-2R, Catalyst Systems, Pune, India	600	5	Glass beaker	Repaglinide	PEG 6000	NA	86 (120 min)	[71]
2013	Catalyst systems, Pune, India	440	NA	NA	Glipizide	PEG 4000	Ice bath	NA	[83]
2014	CATA-2R, Catalyst Systems, Pune, India	560	NA	Glass beaker	Candesartan Cilexetil	PEG 6000; HPMC E5	NA	> 90 (5 min)	[72]
2016	P70F23P-G5(SO), Glanze	550	10	Crucible	IBU	Soluplus^®^	Liquid nitrogen (−196 °C)	NA	[73]
2017	ME0113M1, Samsung	900	Different time of interval	Glass beaker	Mefenamic acid; flufenamic acid	Pluronic F127^®^; Eudragit EPO^®^; PEG 4000; Gelucire 50/13	RT, with one exception under −80 °C	80 (40 min) for optimum prescriptions	[74]
2019	P70F23P-G5(SO), Galz, Guangzhou, China	700	2	Porcelain dish	IND; fenofibrate	Soluplus^®^	Liquid nitrogen at −196 °C	NA	[84]
2020	ME0113M1, Samsung	500	NA	Glass beaker	Luteolin	PEG 4000	NA	>60 (20 min)	[75]

NA: not available.

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
