# Peer review of "Microwave-Induced In Situ Amorphization: A New Strategy for Tackling the Stability Issue of Amorphous Solid Dispersions"

_pharmaceutics, 2020, doi:10.3390/pharmaceutics12070655_

Round 1
Reviewer 1 Report
This review manuscript by Qiang et al. deals with the in situ (either in powder or tablet) amorphisation of crystalline APIs, which has the potential to avoid all the inconveniences related to the physical stability of amorphous solid dispersions during downstream processing and storage. The manuscript is concise and good to read. However, the subject of it does not strike to the reviewer as a "ripe" subject to assemble a review publication about it in spite of its potential in the future. The main part of the manuscript (even the authors stressed that this has the true, practical potential), i.e. the in situ amorphisation in the final dosage form, contains 4 references (22, 46, 47, 84), and all 4 of them can be related to the authors (at least one common author). In my opinion, no comprehensive conclusions can be yet made about this subject. One publication from this year (DOI: 10.3390/molecules25051068) is missing from the manuscript, but that one is also related to the authors.
Reviewer 2 Report
The submitted work certainly presents an interesting and novel concept. However, it is not clear whether the application of MWs will simplify or complicate even more and increase the uncertainty as to the nature of the solid state of the drug in the administered dosage form. This is because as shown in the examples provided by the authors, the success of amorphization depends on uncontrollable environmental conditions such as relative humidity, and optimal settings of the MW appliance, i.e. correct power input and evaporation capability, irradiation time, implying compexities in the application of the proposed strategy. Furthermore, although simple in its use, MW is not risk-excluded considering the harmful effects of any escaping radiation signifying that not a simple house hold MW is suitable. Therefore, the authors must specify the administration set up e.g. hospital care unit and precautions.
On the other hand, the review is very well written and except the Table’s presentation it is nearly ready for publication. Therefore, I suggest resubmission of the manuscript but in a more realistic perspective. Authors should clearly state the limitations of the application of the concept and they should propose MW use only as part of the manufacturing process of the drug product.
Specific points
Abstract
L 16 - How is amorphicity confirmed, this is a serious concern and patient is not the qualified person to provide an answer
Intorduction
L – 43 Please add ‘pharmacologically’ before inert
L – 65 Please add information on the operation characteristics of the oven, e.g. power, temperature control, evaporating capability.
L- 144, 145 – Sentence is not comprehensible. Authors should make clear of what is the state of the drug leaving the manufacturing site, what is the state when it reaches the patient or care unit and what is expected to have after MW processing
Table 1 – The usefulness of this Table is questionable. The reader is more interested in the theme of MW application. If authors insist to keep it then, increase of the width of columns 4 and 6 is needed to avoid awkward line breaks.
L 197 – ‘within the matrix of material’ could be replaced with ‘within the material structure’ since ‘matrix’ in the manuscript also refers to polymeric matrix (L 158)
Table 2 – Please reorganize by re-adjustment of column widths to avoid inconvenient breaks of the descriptions of instrument type and carrier.
Figure 2. Any comments about the ejectability of these compacts from the compression die?
L 432 – Please add ‘of’ before no degradation
L 433 – Please add ‘that’ before was detected
Reviewer 3 Report
The manuscript provides an overview of microwave-induced in situ amorphization approach to circumvent issues with physical stability and downstream processing of amorphous solid dispersions (ASD). The authors firstly point out current issues in the development of ASD in order to introduce the advantages and significance of in situ amorphization before reviewing general principles of microwave heating and bulk preparation of ASD. Then the main part of the manuscript was presented, discussing various aspects of microwave-induced in situ amorphization within the final dosage form.
As ASD technique is attracting more and more attention from both academia and industry to improve the solubility and dissolution of poorly water-soluble drugs, the new technology using microwave to in situ amorphize crystalline drugs (and so this review) would be very interesting for the readers of Pharmaceutics.
In Section “5. Conclusions”, the authors mentioned some facets of microwave-induced in situ amorphization that require further investigations, including
- Underlying mechanism
- Influencing factors
- Chemical stability: drug degradation and hydrolysis
- Selection criteria of excipients: other polymeric and non-polymeric excipients rather than PVP
- Practicality of the dosage form after processing: sample deformation, low porosity
In my opinion, it would be more valuable for the readers if the authors discuss in more details (preferably in a separate section) current challenges as well as future perspectives of microwaved-induced in situ amorphization within the final dosage form.
Some specific topics might be worth further discussing:
1. The microwave device: the microwave oven used in Ref 22, 46, 47 and 84 was equipped with the inverter technology, allowing a real power reduction without pulsing the radiation on and off. How different the amorphization process would be when using common household microwaves without inverter technology and without well-controlled irradiation power? How important is the microwave methods (intermittent vs. continuous exposure to microwave radiation)?
2. Amorphization homogeneity: Doreth et al. (Ref. 22 and 46) found two Tgs after microwave treatment, one for the drug-polymer mixture, the other for the pure polymer, indicating phase separation due to inhomogeneous amorphization. How to control the amorphization uniformity within the tablets? How important is the position of tablets in the microwave oven during amorphization?
3. Possible air bubbles formation due to water evaporation during microwaving that results in tablet deformation – how to tackle?
4. Disintegration problem of microwaved tablets – how to tackle?
5. Would compaction force impact the microwave-induced amorphization capacity?
6. Residual crystals: What is the impact of residual crystals (due to incomplete amorphization) on the supersaturation profile of dissolved drug, especially for fast crystallizing compounds? What kinds of drug compounds this technique would suitable for?
7. Administration: in practice, how to administer this dosage form? How to conveniently and quickly add moisture to the formulations before microwaving? How to microwave the tablets correctly? Would microwaving be done by health care professionals before distributing to the patients or by patients themselves at home prior to administration?
Round 2
Reviewer 1 Report
The authors have made some changes in the manuscript that address the questions/remarks made by the reviewer. Therefore, the manuscript can be accepted for publication.
Reviewer 2 Report
Authors have responded well to the comments of the reviewers and adjusted the perspective of the work in concordance to reality. The paper is now publishable